# Psychosocial Intervention in European Addictive Behaviour Recovery Programmes: A Qualitative Study

**DOI:** 10.3390/healthcare8030268

**Published:** 2020-08-13

**Authors:** Antonio Molina, Jesús Saiz, Francisco Gil, María Luisa Cuenca, Tamara Goldsby

**Affiliations:** 1Department of Social, Work and Differential Psychology, Universidad Complutense de Madrid/UCM, 28223 Pozuelo de Alarcón, Spain; jesus.saiz@psi.ucm.es (J.S.); fgil@psi.ucm.es (F.G.); marialcu@ucm.es (M.L.C.); 2Department of Family Medicine and Public Health, University of California San Diego, La Jolla, CA 92093, USA; tgoldsby@health.ucsd.edu

**Keywords:** addictive behaviours, subgroups of risk, psychosocial intervention, psychosocial factors, recovery, qualitative analysis

## Abstract

Background: The vulnerability of drug users in treatment increases their risk of social exclusion. Psychosocial interventions aim to solve or minimise this risk, increasing social integration for individuals in treatment. In this study, the effectiveness of various European drug recovery programmes was studied by examining psychosocial intervention groups with vulnerable populations. Methods: Research was conducted utilising qualitative methodology in the following manner: bibliographical review (reports and papers) and 18 interviews with key informants (from nine European countries). Results: Treatment programmes were found to be effective for recovery from opiate use, however, social recovery programmes were not found to be effective. For females in treatment and young drug users, the adequacy of programmes received medium-to-low evaluations. Conclusion: It is necessary to increase the effectiveness of psychosocial intervention in recovery-based programmes. In addition, the relevance of these programmes in relation to psychosocial characteristics of the various risk groups (except opioid consumers) should be validated and revisited. Additionally, psychosocial intervention professionals should take a more active role in such programmes, especially in intervention with young people (in health education, prevention, early detection, and intervention) and with women (in regard to the detection and effective intervention for socio-health problems, social and job reintegration, access inequality, and use of services available due to gender).

## 1. Introduction

Addictive behaviour problems can be treated effectively when individual with substance abuse disorders have adequate access to evidence-informed, quality treatment and rehabilitation services [1]. All barriers to treatment services need to be minimised to better meet the needs of people who present problems with addictions [1,2,3,4]. Actual addictive behaviour treatment networks are composed of harm reduction services and recovery-oriented programmes, with approaches based on the biopsychosocial model, defined as integral or holistic treatment because of the intervention in different aspects of the person (biological, psychological, and social topics) [5,6]. These networks are different depending on the policies of any country and the experience with drug epidemics, such as alcohol, opiate, or cocaine problems [1,2,3,4]. Approaches include multidisciplinary teams (physicians, social workers, and psychologist at minimum) applying common protocols and action plans [7]. People with addictive-behaviour-related problems are at higher risk of suffering social exclusion [3,4] and require not only strength-based interventions but also community engagement and pathways that challenge discrimination and social exclusion [8]. There are strong links between poverty, deprivation, widening inequalities, and problem drug use but not in a linear way. It may involve fragile family bonds, psychological discomfort, low job opportunities, and few community resources [8]. A “psychosocial intervention” usually refers to some type of multidisciplinary intervention consistent with theoretical and practical models from Clinical Psychology and Social Psychology. In addition, models from Anthropology, Sociology, Social Pedagogy, Medicine (in recent years, especially Child and Youth Psychiatry), and Social Work, among others may be utilised [9]. Psychosocial intervention may be used in the field of community health promotion [10], direct intervention with risk groups, social values programmes [11], social integration, development of social support networks [12], and in any type of drug or other addiction treatment and prevention programme using a multidisciplinary approach. Its contribution can be important in early detection and intervention, the initial approach, and referral to specialised resources [13]. O’Brien and O’Brien [14] applied principles of normalisation and validation of social role to the design of addiction treatment services, including “Five Fundamental Goals”:Presence in community, so these services are feasible and accessible for the given population.Election, meaning offering consumers the ability to choose the type of intervention they will receive, which allows them to play an active part in the decisions regarding these interventions [15].Competence or having the necessary experience and skills for active participation.Respect.Community participation and active relationships with social context members [16].

This global scheme theoretically allows adaptations to the characteristics of the various risk groups. As the European Monitoring Centre for Drugs and Drug Addiction (EMCDDA) states, there are some groups of drug users with special problems, needs, or vulnerabilities that require specific intervention [3]. Based on this analysis, adapted centres and services have been developed to better respond to the needs of the following groups: older individuals with social health problems and with a problematic consumption of opioids and/or polyconsumption (consumption of several drugs); women who have drug-related problems; vulnerable young people who present problems related to addictive behaviours and other risky behaviours; and families of individuals with problems related to addictive behaviours [3]. Drug treatment policies and intervention practices in addictive behaviours are based on harm reduction, recovery, and sustainable livelihoods [1]. International consensus is clear regarding the need to broaden these programmes using the biopsychosocial perspective [5,6,17], and any type of intervention should include topics such as recovery [18] and social support [12] to be effective in the long term [1].

Studies examining psychosocial support applied to substance abuse treatment are addressed in three dimensions [1]: structural, functional, and contextual. The structural dimension of these studies includes topics such as the size of the social support network, density, reciprocity, and interaction between components; evaluation of structure is based on sociodemographic indicators or social networks analysis. The functional dimension of these studies addresses the emotional and other aspects such as actual support, perceived support, feasibility, and satisfaction with support. Lastly, the contextual dimension is interested in the examination of participants in regard to social support, moment of support (time in their lives when they are obtaining treatment), and participants’ subjective perception. Family support as the primary supporting social structure is fundamental in this topic.

Recovery [18] is an important concept regarding the treatment and rehabilitation of addictive behaviours. It means not merely reducing or eliminating the use of drugs (including alcohol) [19], it also translates into becoming an active member of society [17]. Moreover, it also does not mean a process of “natural recovery” by which the addicted individual terminates the use of drugs [20]. Best et al. [21] define the following three concepts linked to recovery: (1) contagion is the capacity of an individual in recovery to influence his/her social context; (2) connection is the capacity to build community and society; and (3) homophily is a tendency to relate to and bond with individuals that are similar to us.

Granfield and Cloud define “recovery capital” as the combination of internal and external resources that can be drawn to start and sustain recovery from substance abuse problems [22]. There are three phases to recovery capital (RECCAP): (1) scientific assessment of strengths and weaknesses; (2) planning the individual’s care in relation to their strengths; (3) the recovering individual having strong and solid links with groups and activities oriented to recovery.

White and Cloud [23] state the following:

“Recovery capital constitutes the potential antidote for the problems that have long plagued recovery efforts: insufficient motivation to change AOD use, emotional distress, pressure to use within intimate and social relationships, interpersonal conflict, and other situations that pose risks for relapse (…). Strategies that target family and community recovery capital can elevate long-term recovery outcomes as well as elevate the quality of life of individuals and families in long-term recovery”.

For these authors, there are three phases for identification of RECCAP: (1) Support screening and brief intervention (SBI) programmes; (2) assess recovery capital on an ongoing basis; and (3) use recovery capital levels to help determine level-of-care placement decisions. In this sense, Best [24] divides recovery capital into three aspects: personal recovery capital composed by skills and abilities recovered/empowered during rehabilitation process, especially emotional skills; social recovery capital is the impact of recovery on social groups, especially family and social networks; collective recovery capital includes the impact of recovery in the social context, especially cost/benefit balance. In the last ten years, there have been several research studies focused on the relevance of recovery (fundamentally, recovery capital and social recovery) as the basis in addictive behaviour treatment [1,3,24,25,26]. However, these studies utilise general intervention designs, not considering individual situations and special needs [27] of risk patterns for drug abuse in the category “addict in treatment” [3]. The evolution of recovery-oriented programmes to actual interventions has been heterogeneous in model, techniques, and perspectives [28].

## 2. Materials and Methods

### 2.1. Study Design and Procedure

This study draws on qualitative research methodology (with a complementary part of statistics), using an action research approach [28]. It was included in the project “HOME/2014/JDRU/AG/DRUG/7092-Triple R: Rehabilitation for Recovery and Reinsertion” [29], coordinated by San Patrignano about transfer of good practices in recovery (see Figure 1), especially in peer-to-peer learning and social entrepreneurship for people with addiction problems in Europe between five European countries (Spain, Italy, Belgium, Sweden, and Croatia). The study was developed in two stages: (1) bibliographical review and (2) interviews.

For data collection, the Consolidated Criteria for Reporting Qualitative Research/COREG [30] was used, which is a checklist designed for the reporting of qualitative studies. This tool consists of 32 items grouped into three domains: (1) research team and reflexivity; (2) study design; and (3) findings, data analysis, and research report. This technique complements other objectification processes such as saturation or methodological triangulation. Additionally, it assists in keeping opinions and biases out of the data to ensure more objective data in order for the research team to report findings of the study, context, results, analysis, and interpretation. A multiple-triangulation technique was used to classify the information, in order to ensure the reliability of results.

#### Study Aims

The aim of the present study is to assess the adequacy of actual addiction recovery programmes in Europe, according to the health, social, and psychological needs [27] of predefined subgroups of addictive behaviours in treatment [3,25] and highlight the role of psychosocial intervention in these European recovery programmes.

The specific aims of the study were (1) to analyse the adaptation of European recovery-oriented programmes to actual patterns of persons in treatment; (2) to identify what particular profile or groups are most suited to these recovery programmes (including reasons for suitability); and (3) ways to adapt the psychosocial intervention for the recovery of these specific populations.

### 2.2. Measures

#### 2.2.1. Literature Review

A systemic literature review of current reports and research was undertaken. The databases consulted were Medline, Scopus, and Google Academics, using the descriptor “Addiction Recovery”. Selection criteria were the following: (1) indexed and/or peer-reviewed publications, (2) written in a European Union language, (3) published between 2012 and 2019, and (4) with the word “Recovery” in the title of the publication. In total, 205 articles were found, with 18 publications meeting the selection criteria. To check the validity and adequacy of this review, the Preferred Reporting Items for Systematic Reviews and Meta-Analysis (PRISMA) checklist [31] was utilised. This tool is a 27-item checklist that is used in reporting meta-analysis and systematic reviews and also includes a 4-phase flow diagram. All publications used in the research were examined for adequacy and quality by utilising the PRISMA checklist.

#### 2.2.2. Interviews

Following the literature review, semi-structured interviews were conducted. The researcher previously designed an interview script using various questions to link reviewed literature to the study objectives (see the guiding questions in Table 1). In this way, participants can express opinions, clarify answers, and even delve beyond the questions when it is considered useful to data collection [32]. Tagging of informants for coding was done with the combination Gender–Position–Nation. Data analysis was done with grounded analysis approach. The transcript data were grouped and reanalysed by theme according to the group of drug users at risk under treatment [3]. The grouped data were then disaggregated according to four overarching categories into “Recovery” concepts: sociodemographic characteristics, treatment network, psychosocial support, and recovery pathway.

To complete the data, a brief statistical analysis has been done (percentages and means) from the answers collected in the interviews.

### 2.3. Sample

Participants were selected due to their professional experience in the field of policy making, prevention and/or intervention in addictive behaviours, and academic research or studies on the subject [33]. The interviews were carried out via videoconference or in person by the first author, and lasted between 45 and 60 min. The interviews were conducted in English, Spanish and Italian, depending on the participant’s linguistic preference. (See Table 2).

All the 18 participants had been recognised as key experts at the European level. The participants were located in Belgium, Croatia, Italy, Ireland, Netherlands, Norway, Russia, Spain, and Sweden. Eight participants were females. Participants were over 40 years old, with more than 10 years of professional experience and specific formation in drug treatment/research (Master and/or Doctor degree). The participants were representative of the following groups: physical health; psychotherapy; research/teaching; direct intervention; legal aspects; and policymakers. For this study, we selected only international experts and not professional staff members or treatment users to decrease the influence on “reaction rhetoric” and the biases of effected groups of treatment stakeholders [34].

One interview was conducted with each expert. Written informed consent was obtained from all participants, with a guarantee of confidentiality and anonymity. This study received approval from the UCM Deontological Research Commission, ethical approved project identification code is UCM PR2019_20_043. Multiple types of data were collected during interviews, including audiotapes of the participants, notes taken by the moderator and assistant moderator, items recalled by the moderator, and field observations when the interview was conducted in person.

To avoid possible researcher bias of interviews, weekly supervision meetings were held with the participation of the entire staff. Reliability among judges was calculated with seven randomly chosen interviews (35.7%) at the end of data collection, coded by two researchers. The reliability among judges was quite good, with a Kappa value of between 0.75 and 0.82 and an average of 0.79. The SPSS Statistics V23.0 programme (SPSS Inc., Chicago, IL, USA) was used to calculate the Kappa value. Qualitative data analysis was conducted using Atlas.ti-8 software.

## 3. Results

Results are presented below and grouped into specific subgroups (see Table 3) of the drug abuse population [6], accompanied by various verbatim responses from participants.

### 3.1. Older Individuals with Social Health Problems and with a Problematic Consumption of Opioids and/or Polyconsumption

For the question, “*What are the groups who fare less well? Opiate users older than 40 years, women with addictive behaviour problems, or young users at risk?*”: In the opinion of 83.3% of the informants, recovery programmes are most suitable for people in treatment for opioid use or polyconsumption of drugs. Opioid addiction or polyconsumption of opioids plus additional drugs is a situation that has remained stable and problematic since the heroin epidemic that affected Europe in the 1980s and 1990s.

“*The treatment available for opioid users over the age of 40 has not changed and has remained the same since the 1980s: opioid substitute treatments. At the time, there was only methadone available, now they also have buprenorphine. There is also slightly more attention to this health situation now (the eradication of HIV and HCV has been implemented for active drug users) and to their social needs. On the one hand, this is a way to partially improve their quality of life. On the other hand, this chronic stability often prevents them from seeking recovery.*” (Female, Professor of Public Health).

Regarding the question, “*How has the adaptation of actual recovery programmes been to opiate users older than 40 years?*”: 66.6% of study participants’ perception is that, for this group of addicted individuals, there are sufficient, efficient treatment programmes, especially programmes that address socio-health and psychosocial issues on a regular basis and whose operation is validated.

In the opinion of another study participant, “The services and health needs of people addicted to opioids such as detoxification and OTS are widely available and easily accessible.” (Male, Clinical Director).

For the questions, “*What are unmet needs which are linked to poor pathways?” and “Which specific needs do you think these groups have?*”: 77.7% of informants express that in recovery-based programmes, there are aspects that are addressed very intensively and effectively, such as self-care behaviours, behavioural habits, cognitive behavioural analysis of addiction, and relapse prevention. However, there are other important treatment programme aspects that are not integrated into the programmes. As one participant said, “*substitute treatments are resolved (used effectively). The problems of where to live are not solved. The threat problems of the trafficking gangs are unresolved. The problems of the needs of women with children are unresolved. Trauma and mental illness are not being addressed.*” (Female, Programmes Manager).

For the question “How has the adaptation of actual recovery programmes been to opiate users older than 40 years?”: According to 44.4% of the participants in the present study, various types of opiate users over 40 years of age are not properly served by the intervention networks.

One female participant, whose occupation is Professor of Public Health stated, “*Opioid addicts have specific treatments. Youth and women addicted to stimulants, amphetamines, and cannabis do not receive appropriate treatment. If we go further, programmes appropriate to gender and age do not exist.*”

In answer to the question, “*Have you identified any areas of need among your target population that practitioners were largely unaware of?*”: 72.2% of participants identified social stigma and difficulties of complete integration, especially job placement, as primary problems in obtaining treatment. These issues appear to go hand-in-hand, as well. As one informant indicates, “*levels of stigma and marginalization remain very high in society.*” (Female, Communication and Drug Policies coordinator).

### 3.2. Women Who Have Drug-Related Problems

In response to the interviewer’s question regarding “*What are the groups who fare less well? Opiate user older than 40 years, women with addictive behaviours problems, or young users at risk?*”, another problematic category was discovered, that of lack of care for females. Care for women with addiction problems in the European network appears to be extremely lacking. The need for programmes to be gender sensitive in order to be more effective seems clear; however, specific interventions with a gender focus appear to be sorely lacking in the public health field as well as in social and psychological intervention.

One male study participant, whose occupation was clinical coordinator, stated, “*Treatment for women with substance abuse problems should take into account not only biological aspects, but also social and environmental factors, trauma, and sexual assault (which is more common in women seeking treatment for drug dependence problems) and related factors with financial independence, pregnancy, and childcare.*”

For the question, “*How has the adaptation of actual recovery programmes been to women with addictive behaviour problems?*”, there is clearly a need for training mental health professionals in gender sensitivity (77.7% of informants). This is needed in both the design and execution of treatments, as well as in the care of women who attend social health centres, in order to avoid gender stereotyping and to include these mandates in the intervention.

One female participant, whose occupation was European projects coordinator stated, “*There should be training programmes for professionals to recognise the symptoms of drug and alcohol abuse in women, especially for older women. Above all, the medical team, psychologists, and liaison nurses should be trained in how different patients react to pain relievers depending on whether they use drugs or not.*”

The question was asked, “*Which are the specific needs these groups have: need for treatment, care, and aftercare relating to drug use?*”. This situation becomes even more complex when one considers that the profiles of women with addiction problems are diverse and that they go beyond the existing social identity of the unique and homogeneous profile of the female addict. The various profiles of females with addiction problems may include differing behaviour patterns, family responsibilities, and habitual drug use combined with psychological disturbances such as anxiety and depression. This situation may lead to chronic problems and may also contribute to their invisibility in society. As one female participant, a psychologist, indicates, “*there is also a large group of highly functional, working, and married women who have children and who are addicted to stimulants, cannabis, and alcohol, who after years of addiction experience anxiety, depression, and insomnia.*”

The following question was asked: “*Have you identified any areas of need among your target population that practitioners were largely unaware of?*”. In considering this question, it is important to address the accessibility of treatment for underserved females, taking into account factors such as psychosocial risks. Moreover, it is vital to increase the visibility of the treatment programmes to make women with addiction issues in more vulnerable situations more aware of these programmes, and to take into account barriers to care access, such as fear of the legal consequences regarding custody of the children. As one female participant whose occupation was harm-reduction programmes coordinator states, “*I think those unaware of the services and programmes are even more chaotic and at higher risk. Perhaps there are more women hidden from services because of possible problems with children.*”

Participants were also asked the following question: “*Have you identified any areas of need among your target population that practitioners were largely unaware of?*”. In considering this issue, it is important to address the psychosocial problems related to addictive behaviours in women, especially gender-based violence in couples and sexual violence. Many of these women need some defence or protection from the abusive partner, so the intervention needs to address this problem, if it is to be effective.

One female participant whose occupation was Professor of Public Health stated: “*There is a need for more attention for women. They appear to have no access to information about the chances of recovery. They are also victims of violence, gender, and sexual exploitation. Systematic efforts are needed to address the issues related to childcare, pregnancy, and trauma that they often experience. Continuity of care, support after treatment, and resocialisation are not recognised as key elements in recovery.*”

Participants were asked for their opinions regarding the specific shortcomings of recovery-based treatment programmes. It was discovered from their responses that access to accommodation for the individual with addiction issues (83.3%) and socio-occupational reintegration (89.9%) were considered global shortages in European countries. In addition, they stated that other gaps in the system included problems with perceived social support and issues with collaborations between public organisations (61.1%), public care networks (mental health, violence, and substance abuse) (77.7%), and private care. They are considered important aspects of these programmes that are based on empowerment of the individual with addiction issues and assisting the individual in building personal and social competencies.

One female participant, a Communication and Drug Policies coordinator stated: “*In my experience, there are unmet needs in employment and residence in the first place, and collaboration between local authorities and private employment providers should be supported by government authorities.*” In the opinion of another expert, “*…the needs related to continuity of care remain unmet: social insertion, activities with the meaning of social support, work, and residence.*” (Male, European Projects Coordinator).

Study participants recognised that the gender issue is an important one in determining the suitability of treatments currently being discussed (94.4%). It seems appropriate to implement more specific programmes and interventions, especially by professionals who have the most direct contact with people undergoing treatment. As one study participant said, “*we have a very low number of women asking for help, and programmes with additional gender sensitivity are generally not available to them.*” (Female, European Projects Coordinator). Another female study participant, a psychologist, stated: “*Gender issues are now central to the treatment discussion, and more programmes that take into account the specific needs of women are now available in some public drug dependency services. Professionals are now more careful that women have different health problems, as well as the existence of aspects related to children and pregnancy, and are slowly bringing these concerns closer to programmes to make them available to women.*”

### 3.3. Vulnerable Young People Who Have Problems Related to Addictive Behaviours and Other Risky Behaviours

Participants were asked, “*What are the groups who fare less well? Opiate users older than 40 years, women with addictive behaviours problems, or young users at risk?*”. There appears to be a consensus (94.4% of participants) that among young people who currently have problems related to addictions: care is inadequate, not adapted to consumption patterns, and psychosocial and other problems of the 21st century, and recovery programmes are not effective venues for intervention. In the opinion of one study participant, “*young men do less well than others and men do less well than women.*” (Male, Professor of Psychology).

When our experts were asked, “*What are the differences between the profiles of drug users engaged in recovery treatment now compared to those known in the past or to those unknown to services?*”, there seemed to be a consensus (88.9%) regarding the problem. Their answers revealed that the primary issue is that the change in consumption patterns and priorities of this age group has not been addressed in the approach strategy of recovery-based programmes.

“*There has been a change in the profile of people seeking help for their addiction. Now we have young people, 18–24 years old, of both sexes, who use amphetamines and cannabis problematically, and experience moderate-to-severe mental health consequences. They are very aggressive, involved in violent and criminal activities, without strong support from families, who are now realising the signs of the addiction problems earlier compared to the period 10 years ago, when addiction runs started 10–15 years before the family got involved.*” (Male, Professor of Criminology).

Participants were asked, “*How has been the adaptation of actual recovery programmes been to young users at risk?*”. It was revealed from this question that the expert participants in the study were especially concerned (83.3%) that the programmes have not adapted to challenges surrounding young people who present risky behaviours and who are still functional, on the border of what can be considered an addiction, and who cannot find programmes suitable to their profiles and needs.

One female participant, whose occupation was European Projects Coordinator stated, “*We have young users of high-functioning cannabis derivatives, amphetamines, and stimulants who cannot find adequate support within the system or do not believe they need help and come to us because of the family.*”

There appears to have been proposals for action to address this issue; however, the proposals seem to have been more of the reactive type than the prospective type. Participants were asked, “*What are unmet needs which are linked to poor pathways?*”. It was revealed from their answers that another of the problems considered is the reduced use of health education and early intervention for these young people, which aggravates the severity of the problems and their consequences.

As one male participant, a Professor of Criminology said, “*The increase in the number of young people with serious addictive behaviour, and the decrease in the age of first contact with drugs, sensitised those responsible for the services and also stimulated the application of prevention initiatives aimed at children and adolescents.*”

Participants were asked, “*Which are the specific needs these groups have: need for treatment, care, and aftercare relating to drug abuse?*”. Our experts indicated that a coordinated and comprehensive approach utilising numerous resources and professionals in adapting the programmes is necessary, especially in the case of treating young people. In addition, it is important to include social support strategies (66.6%) and contingency management training for social groups (77.7%) in these programmes, to improve the prognosis and effectiveness of the programmes.

Another male study participant, a Professor of Criminology stated, “*Young users are sensitive to social cues, with peer groups and families being very influential. Therefore, the specific needs of this population are related to the need for improvements in services that facilitate parental involvement, integrate school, and recognise the importance of prosocial peer relationships.*”

## 4. Discussion

Psychosocial intervention in addictive behaviour recovery programmes is strongly linked to medical treatment [20]. In the interviews, a significant concern was discovered regarding the coverage of patients with opioid problems older than 40 years of age, especially from a socio-health perspective. Pharmacotherapy substitution treatment is universal in European countries, as recovery programmes are not feasible for all persons with addictive behaviour problems in Europe [25].

Adaptation of recovery programmes to actual patterns and styles of drug abuse and addictive behaviour problems seems to be very limited and still connected with heroine recovery treatments. In the group of opiate users over forty years of age, the degree and type of psychosocial intervention seems to be more delineated than for younger individuals [35], which is reflected in previous studies [36]. However, treatment for other populations, such as women with addiction problems and women with children remains limited [37]. The abundant evidence and the reality observed in these programmes indicate the need to change the current perspective regarding gender as an additional (perhaps unimportant) topic to one where it is considered necessary and important. This allows the creation of more effective programmes for men, women, other gender identities, and lesbian, gay, bisexual, transgender, and intersex /LGTBI groups [13,35,36,37].

In addition, problems exist related to social integration; the stigma of addiction [38] is still quite apparent for these populations, as well as sexual abuse and sexual exploitation [35,36,37,38]. Additionally, interviews revealed that treatment approaches for trauma and mental illness for drug addicts continue to appear ineffective and insufficient. There is a general perception (with the exception of informants from Sweden, Belgium, and Italy) of too few residential programmes for the entire population. In relation to employment and social support, informants of Sweden and Italy present specific programmes with sufficient material and human resources, using networks of public bodies and private companies, as well as social companies developed by recognised organisations (Basta and San Patrignano), seeking the employability of the participants in said programmes. Finally, there are only limited recovery treatment programmes with an evaluation model based on monitoring and follow-up [35,39].

Moreover, childcare and improving parenting skills and competencies are additional treatment programme issues that need to be addressed [14]. There are programmes that are already working with these problems in a standardised way, such as Proyecto Hombre biopsychosocial model [5] and HERMESS—human centered, empowerment aimed, reintegration oriented, motivational driven, educational embedded, self-sustainability focused, social need oriented [29], developed to be references for new recovery-oriented programmes; another standard is “CHIME Model: Framework of elements of psychosocial support for personal recovery” [40]. This model is based on perceived social support and the degree of usage of available resources and capacities. CHIME is the acronym for the various resources included in the model:Connectedness (Connection and social support networks).Hope (Hope and motivation).Identity (Social and personal identity).Meaning (Meaning that the person gives to the social support network).Empowerment (Empowerment; personal and social skills).

These health promotion models for the “recovery cities” are already being utilised in cities such as Goteborg and Stockholm (Sweden) in reducing risk situations due to drug use and associated problems (especially crime and socio-health emergencies), as well as to improve coexistence and citizen participation [21].

The coordination between the health system (especially pharmacological treatments and substitutes, as well as medical protocols) and other services that participate in social intervention in addictive behaviours is another aspect for which attention needs to be paid [36,41]. This coordination, which is perceived as very beneficial and which clearly improves the efficiency of networks and services, requires professionals who are experienced with working in multidisciplinary situations [5].

Another aspect that is evidenced in the present study is the need to include continuous improvement strategies, evaluation, and proposals for action within recovery-based programmes [41]. If not carried out, there will continue to be programmes for particular populations of opioid users and other patterns of drug abuse that are highly effective whilst certain populations such as women and young people will continue to find that treatment is inadequate [38].

### Limitations of the Study

As limitations of the study, even though the sample of the study is composed of experts with high international relevance, it would only be possible to fulfil the qualitative perspective with opinions of other stakeholders (users, families…). On the other hand, qualitative research strategies not only check search-based hypotheses but also generate new lines and deeper descriptive content development [28,32,33]. That is why there is a need to design and implement new recovery programmes using experimental (or quasi-experimental) studies for the monitoring and follow-up of processes and results into programmes of addictive behaviour recovery.

## 5. Conclusions

The participation of several disciplines and perspectives is necessary to guarantee psychosocial intervention in addictive behaviour recovery [24]. However, professionals (social workers, psychologists, social educators, sociologists, nurses, doctors) who intervene with people with addiction problems need to have additional training in cross-disciplinary aspects, which helps to better detect the problem and design more effective and fair interventions [39]. It is necessary to redefine the objectives of the psychosocial intervention for people with addiction problems, promoting the use of strategies based in multidisciplinary action that allows the development of a more active role for the beneficiaries of the programmes, such as person-centred planning [14] or the model “Power–Threat–Meaning Framework [27].

Recovery-oriented programs must be integrated and connected with harm-reduction networks, social services, health system, and employment service. Recovery models have to integrate elements such as social support, active participation, gender perspective, and social integration. It is also necessary to create specific actions for several collectives, such as developing evaluation systems to validate the efficiency and adequacy of recovery-oriented programmes. An adequate socio-health and psychosocial intervention is important to reduce these problems [26,28,41].

## Figures and Tables

**Figure 1 healthcare-08-00268-f001:**
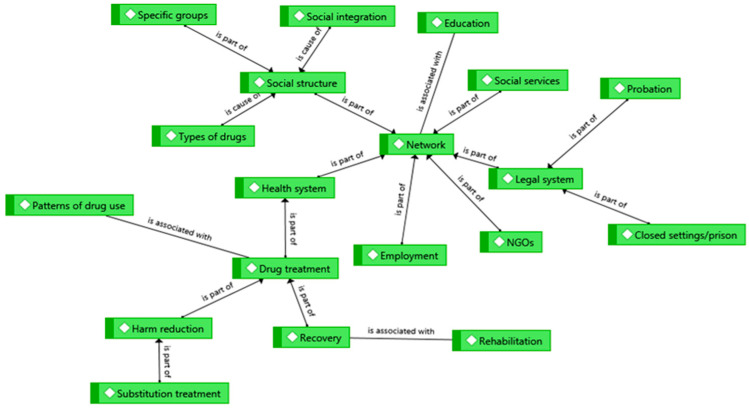
Triple R (Rehabilitation for Recovery and Reinsertion) categories network.

**Table 1 healthcare-08-00268-t001:** Questions for “Adaptation of recovery programmes to European Monitoring Centre for Drugs and Drug Addiction (EMCDDA) vulnerable subgroups” study.

What are the differences between the profiles of drug users engaged in recovery treatment now compared to those known in the past or to those unknown to services? How does the profile of drug users engaged in recovery treatment vary by drug, gender, ethnicity, age, referral source, and housing status? What are the groups who fare less well (such as opiate users older than 40 years, women with addictive behaviour problems, or young users at risk)? How has the adaptation of actual recovery programmes been for opiate users older than 40 years? How has the adaptation of actual recovery programmes been for women with addictive behaviour problems? How has the adaptation of actual recovery programmes been for young users at risk? What is the balance of life of individuals on a recovery pathway? Is this balance consistent with ambitions for recovery? What are the unmet client needs that are linked to poor recovery pathways? Which specific needs do you think these groups have? Have you identified any areas of need among your target population that practitioners were largely unaware of? Which of the needs of your target population are currently being met and which are not being met? What are the implications for families of persons with addictive behaviour problems? What kind of services do you offer to families of persons with addictive behaviour problems?

**Table 2 healthcare-08-00268-t002:** Study participants.

Participants	Profession
Female	Professor of Public Health, Ireland
Female	Programmes Manager, Sweden
Female	European projects coordinator, Italy
Male	Clinical Director, Croatia
Female	Communication and Drug Policies Coordinator, Serbia
Female	Psychologist, Spain
Male	Clinical Coordinator, Croatia
Male	Professor of Criminology, United Kingdom
Male	Professor of Criminology, Croatia
Male	European Parliament
Male	President of European Network
Male	President of European Network
Male	Professor of Psychology, Belgium
Female	International Projects, Russia
Female	European Projects Coordinator, Sweden
Male	European Projects Coordinator, Italy
Male	Treatment network coordinator, Belgium
Female	Harm-Reduction Programmes Coordinator, Croatia

**Table 3 healthcare-08-00268-t003:** Main dimensions and categories in the study.

Sociodemographic Characteristics	Treatment Network	Psychosocial Support	Recovery Pathway
Gender	Health system services	Families	Drug/addictive behaviours
Age	Social system services	Social networks/friends	Life quality/health
Level of studies	Mental health services	Social participation	Stigma
Economic level	Harm-reduction services	Perceived social support	Relapse/Prevention
	Recovery services	House/living places	Recovery capital

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
