# Peer review of "Psychosocial Intervention in European Addictive Behaviour Recovery Programmes: A Qualitative Study"

_healthcare, 2020, doi:10.3390/healthcare8030268_

Round 1
Reviewer 1 Report
Dear authors,
This looks like an interesting study however the current structure and presentation of results are not fit for publication.
Introduction
- Your introduction does not explain why social exclusion is a problem, what a multidisciplinary team approach is or what psycho-social support is.
- There is an over reliance of verbatim quotes from quoted sources and bullet points. These are not consistent with academic standards.
- The introduction overall was very confusing and after reading it twice i was still unclear.
Methodology
This section is full of unnecessary information like the preamble on the consolidated reporting framework.
You need to provide an overview of the Triple R project in the introduction and clearly explain how this project is nested in it.
Is your literature review a systemic review or did you refer to those guides. There are standards for reporting this information please be clear.
You do not list the type of qualitative methodological approach you took. You do not explain how the data was coded and analysed.
The 'judging' of coded interviews does not make sense.
Why and how do you privilege so called experts above other groups?
For this study we selected only international experts and not professional staff 182 members or treatment users to decrease the influence on ‘reaction rhetoric’ and the biases of effected 183 groups of treatment stakeholders (31).
Results
These are not formatted in accordance with qualitative academic traditions. As you have not stipulated the analysis you have done it is impossible to provide further feedback.
The results include analysis:
'Presumably, the low number of individuals with addiction problems in treatment programmes affects the adequacy of said programmes'.
Discussion
Given i am not clear on the introduction or the results it is hard to comment on the discussion.
It does however lack a section that looks at specific strengths and limitations
Conclusion
Basic rule of a conclusion is no new information. Please revise and align to academic standards.
It needs a serious revision with focus on clarity.
Author Response
A/A Reviewer 1.
In attached file you can find our answers to your comments and suggestions. If you have got any kind of doubt or comment, do not hesitate to contact us. Lot of thank for your time and efforts. Kindly.

Reviewer 2 Report
Additional Comments for Psychosocial Intervention in European Addictive Behaviour Recovery Programmes: A Qualitative Study
- Introduction:
- Strengths: Authors did a nice job of providing specific information about psychosocial interventions and addiction recovery. I appreciated the direct quotes as well as detailed information provided about each concept. This allows readers who have little to no background in addiction recovery to be able to understand the concepts that will be explored in this paper. I think this allows for a broader audience to benefit from reading this study.
- Weaknesses: This may be a difference between Europe and the United States, but there has been a move in the last decade (at least in the United States) to refer to addiction as substance use disorders and change the terminology of “addict” to “individual with a substance use disorder.” I would like to see further clarification in the introduction about what is specifically being targeted, as “addictive behaviours” could include (or be thought by the readers to include) groups like individuals with gambling addictions, eating disorders, porn addictions, etc. Is this specifically looking at substance use disorders, or are these other groups included when looking at the treatment programs through the qualitative interviews? This is clarified more in the Materials and Methods section, but it is not clear to me from your introduction where you all are going. Please clarify this from the start.
- Materials and Methods:
- Strengths: I appreciated the time it took to base the interviews on the research collected during the literature review. The detail provided was helpful in that I had a clear understanding how the data were collected and analyzed.
- Weaknesses: I am curious as to why the period of 2012-2019 was selected for the literature review as opposed to something different (like 2010-2019, etc). For the question “What are the groups who fare less well?,” were the interviewees provided with what was in parentheses after (“Such as opiate users older than 40 years..”)? It seems to me that this would bias the responses to choosing one of those three categories as opposed to getting a free response about which group they find fares less well. Although you provided information as to why you didn’t select professional staff members/treatment users, I still believe that this would be a valuable group to do a similar study with as they are on the frontlines of what is actually occurring in the treatment of individuals with substance use disorders.
- Results:
- Strengths: I appreciated the direct quotes from the interviewees.
- Weaknesses: I found some of the responses interesting, in that they differed from my experience working in the treatment of individuals with substance use disorders. I could see if treatment is being conceptualized as medication-assisted treatment, the responses that opioid users have the best outcomes and most access make sense. However, many programs (at least in the United States) are appropriate for any substance use disorders and include psychosocial interventions, mental health treatments, and trauma treatment. I would have liked to see this further clarified somewhere on what treatments are being thought of by the interviewees and what types of treatment programs are being currently offered in Europe.
- Discussion:
- Strengths: Authors did a nice job reviewing the information gathered during the qualitative interviews and highlighting the areas that will need to be further evaluated, studied, and integrated in the treatment of substance use disorders with various populations in Europe.
- Weaknesses: Did not see anything that I would want to change in the Discussion or needed further clarification.
- Conclusion:
- Stengths: Again, nice job providing an overall summary of findings and future directions based on information gathered in this study.
- Weaknesses: It would be helpful to maybe provide some examples of who these professionals are when saying “psychosocial and socio-health intervention professionals.” Would these be social workers, psychologists, sociologists?
Author Response
A/A Reviewer 2.
In attached file you can find our answers to your comments and suggestions. If you have got any kind of doubt or comment, do not hesitate to contact us. Lot of thank for your time and efforts. Kindly.

Reviewer 3 Report
This article tries to investigate the effectiveness of various European drug recovery programs. Overall, I found the manuscript informative. The introduction contains sufficient explanations of the research background, and objectives are clearly stated.
I have some suggestions below.
(1) Lines 139-143: Overall, the objectives of this study are clearly explained. I wish to invite the authors to consider this part as the last paragraph of the introduction. It might be a good idea to make a mixture of these lines with the phrases in lines 119-122.
(2) I believe that the method for conducting the survey needs to be more scientifically reliable. I suggest revising the questionary. Questions should be reasonably scored or become in multiply answer format. In the next step, the results can be statistically analyzed.
(3) Considering the scale of the objectives of the paper, I think the number of samples is too small.
(4) More detailed socio-demographic information about the samples should be provided.
(5) Results are described in a completely comprehensive way, without any clear indicator or applying any sound statistical analysis. The authors have worked hard to conduct this research. However, currently presented results do not look to be much trustworthy from a scientific point of view. Please try to support your objectives with clearly quantified data through a suitable statistical analysis.
(6) Appendix A is suggested to be presented as a separate "supplementary" file.
Author Response
A/A Reviewer 3.
In attached file you can find our answers to your comments and suggestions. If you have got any kind of doubt or comment, do not hesitate to contact us. Lot of thank for your time and efforts. Kindly.

Round 2
Reviewer 1 Report
Hi again,
Few remaining issues that need to be rectified before this is in keeping with academic standards.
Your introduction is still not very clear. I feel you need a few sentences orientating the reader to your research area before dropping into the details of models and networks. Step back and give the wide lens before zooming in. This will greatly enhance your manuscript.
Also there are large terms that are introduced and not explained (biopsychosocial).
The use of bullet points and large quote in an introduction is very unusual, consider revising.
Your conclusion is very long and should not contain references. Please revise in accordance with academic standards.
Language remains a concern throughout the document, some sentences miss words. Your first line of conclusion for instance do not make sense. The drug programs- the ones you reviewed? Recovery is inconsistently capitalised.
Author Response
Few remaining issues that need to be rectified before this is in keeping with academic standards.
Your introduction is still not very clear. I feel you need a few sentences orientating the reader to your research area before dropping into the details of models and networks. Step back and give the wide lens before zooming in. This will greatly enhance your manuscript.
Also there are large terms that are introduced and not explained (biopsychosocial).
The use of bullet points and large quote in an introduction is very unusual, consider revising.
Your conclusion is very long and should not contain references. Please revise in accordance with academic standards.
Language remains a concern throughout the document, some sentences miss words. Your first line of conclusion for instance do not make sense. The drug programs- the ones you reviewed? Recovery is inconsistently capitalised.
Response to Reviewer 1 Comments
Point 1: Introduction
- Your introduction is still not very clear. I feel you need a few sentences orientating the reader to your research area before dropping into the details of models and networks. Step back and give the wide lens before zooming in. This will greatly enhance your manuscript.
Response 1: Thank you for your constructive advice for improving our study. Following the reviewer recommendations, we have introduced in the Introduction section sentences for orientation (line 33-39).
- Also there are large terms that are introduced and not explained (biopsychosocial).
Response 2: Thank you so much for detecting and helping with these errors. Following the reviewer recommendations, we have explained the concept (line 38-39).
- The use of bullet points and large quote in an introduction is very unusual, consider revising.
Response 3: Thank you for your comment. Following the reviewer recommendations, we have redacted again the introduction overall to make it clearer, avoiding bullet point and reducing large quotes (line 76-80; 125-128).
Point 5: Conclusion
- Your conclusion is very long and should not contain references. Please revise in accordance with academic standards.
Response 4: Thank you for your comment about our conclusion. We have reduced the conclusion and increased the references (line 470-486).
- Language remains a concern throughout the document, some sentences miss words. Your first line of conclusion for instance do not make sense. The drug programs- the ones you reviewed? Recovery is inconsistently capitalised.
Response 5: Thank you for your comment about our conclusion. We have redacted again first line and the section to improve the clarity of the conclusion (line 470-471).
Reviewer 3 Report
The authors have made substantial improvements to this manuscript.
Author Response
Response 1: We are very grateful for the great impression that our revision made to reviewer 3. Lot of thanks again for all your comments and proposals that helped us to make a better paper. We went through the text again and, considering also reviewer 1, we made minor typo-like corrections.